# Exploration of Li-Organic Batteries Using Hexaphyrin as an Active Cathode Material

**DOI:** 10.3390/molecules24132433

**Published:** 2019-07-02

**Authors:** Ji-Young Shin, Zhongyue Zhang, Kunio Awaga, Hiroshi Shinokubo

**Affiliations:** 1Department of Molecular and Macromolecular Chemistry, Graduate School of Engineering, Nagoya University, Furo-cho, Chikusa-ku, Nagoya 464-8603, Japan; 2Department of Chemistry, Graduate School of Science, Nagoya University, Furo-cho, Chikusa-ku, Nagoya 464-8603, Japan

**Keywords:** Li-organic battery, hexaphyrin, electrochemical oxidation

## Abstract

Lithium-collaborating organic batteries (Li-[28]hexs) were investigated with [28]hexaphyrin(1.1.1.1.1.1) as an active electrode material. Each hexaphyrin of [28]Hex cathode ideally involved four electrons per unit cycle and performed a typical charge/discharge processes of Li-organic battery. Li-[28]Hex batteries set with fast charging rates showed reasonably stable charge and discharge performances over 200 cycles even though it caused incomplete (2~3 electrons) charge/discharge cycles due to failing the complete charging process. UV absorption changes of [28]hexaphyrin in CH_2_Cl_2_ were supplementary for the electrochemical oxidation, which performed a conversion from [28]hexaphyrin to [26]hexaphyrin.

## 1. Introduction

There is no doubt that the sources of our energy consumptions are essential and dictate the quality of our lives, thus the investigation of electrical power suppliers such as portable batteries have been crucial. Up to now, portable battery systems have been convenient energy suppliers and various types of batteries have been proposed and developed substantially [1,2,3,4,5,6]. Li-organic batteries have also been extensively investigated by adopting various suitable organic molecules [7,8]. To build efficient batteries, we have to comprehend the chemistries occurring in the electronic systems. Molecules that are fully reversible with multiple oxidation states are great candidates for building an efficient battery system.

In general, antiaromatic compounds exhibit unstable behaviors, and are easy to mutate and distort to bear better-stabilized conformations [9,10,11], which is a disadvantage of antiaromaticity. In contrast, norcorrole molecule stabilizes its antiaromaticity successfully by tetradentate–metal ligation with Ni(II). The Ni(II) norcorrole (NiNC) precisely projected advantageous natures of antiaromaticity, such as easy redox processes with a small HOMO–LUMO gap [12]. The efficient redox interconversions between the reasonably stable antiaromatic state and the stable aromatic state were promoted for organic batteries. Norcorrole batteries have shown significantly stable charge/discharge performances in both lithium-associating and lithium-free conditions. Our recent research on norcorrole has derived useful chemistries for proposing better battery systems [13,14]. Those batteries exhibited significantly stable behaviors to supply high capacities over 200 cycles. The next candidates that can deliver a consistent chemistry with NiNC were then scanned to achieve efficient organic battery systems with redox interconversions, which are fully reversible and chemically stable.

Extended π-electron conjugations, like expanded porphyrinoids, can be useful supplies since many of them have shown interesting chemical/physical properties capable of a broad range of practical applications [15,16]. M. Faraday accounted for the chemical manifestations of aromaticity of benzene in 1825 [17]. E. Heilbronner first predicted the existence of Möbius aromaticity, in 1964 [18], which was associated with nonplanar expanded porphyrinoids. Their elongated π-conjugations exhibited unique dynamics for aromatic/antiaromatic phenomena, in external magnetic fields by interrelating with their conformation flexibilities [19,20,21]. The dynamic interconversion was ideally accomplished following Hückel magnetic distributions and paratropic ring currents of turcasarin as reported by J. L. Sessler [22]. Dynamical conversions between Hückel and Möbius aromaticities of expanded porphyrinoids have also been investigated. The magnetic dynamics of benzihexaphyrin was reported by L. Latos-Grażyński [23,24]. A. Osuka and D. Kim have also investigated competing unique magnetic dynamics that happened between the Möbius aromaticities and aromaticities of expanded porphyrinoids [25,26,27,28].

The simplest custom expanded porphyrinoids that can be prepared in convenient synthetic methods are *meso*-aryl expanded porphyrins. Hexaphyrins(1.1.1.1.1.1) that have six repeating units of pyrrole and *meso*-methine groups have been deliberated as the most stable expanded porphyrin of the series [29]. The hexaphyrins exhibit various interesting dynamics in redox reactions and their electronic structures: [26]hexaphyrin(1.1.1.1.1.1) ([26]hex) and [28]hexaphyrin(1.1.1.1.1.1) ([28]Hex) exhibit aromaticity/antiaromaticity in accordance with their respective 26/28π–electronic circuits (Figure 1) [29,30]. Even though the capacity is not that high owing to the large mass value of hexaphyrin, the antiaromatic hexaphyrin electrode was expected to have a related chemistry with NiNC batteries. Li-organic batteries were prepared with [28]hexaphyrin(1.1.1.1.1.1) active electrode and the battery proficiency was investigated. The detail behaviors are reported herein.

## 2. Results and Discussion

### 2.1. Preparation of Li-[28]Hexaphyrin Batteries and Investigation of the Battery Behaviors

*meso*-Aryl hexaphyins have been isolated in two oxidation states, exhibiting two distinct conjugate systems, 26 and 28 π-electrons, whose conjugation was converted to the other by chemical redox-reactions (Figure 1) [24,30]. Hexaphyrin having a 26 π-electrons conjugation path ([26]hex) presented purple in color, in regular organic solvents. By contrast, hexaphyrin having a 28 electrons conjugation path ([28]Hex) showed a dark blue color.

[28]Hex was prepared by following the literature method [31] and was merged in battery electrodes. Composite electrodes of 10% and 20% were prepared by the corresponding process, blending [28]Hex, carbon black (CB), and polyvinylidene fluoride (PVDF) ([28]Hex/PVDF/CB = 1:3:6 and 1:1:3 for 10% and 20%, respectively), pressing the paste, and molding the plate. The electrode was then fabricated with a Li electrode (Li-[28]Hex battery, Figure 2). Cyclic voltammetry (CV) of the battery (Ewe/V versus Li/Li^+^) was examined in a window range of 2.0–4.0 V (Figure 3). The initial range was set high and the high voltage (4.5 V) caused decomposition of the electrode and plotted gradual reduction of the redox curves in the cyclic voltammogram of the battery (Appendix A). The redox curves turn out reversibly stable in an adjusted potential range (2.0–4.0 V), which involved overall 4 electrons (2 + 2 electrons).

Battery performances with the same current window involved 3~4 electrons (Figure 4). The unit capacity is relatively smaller (23.52 A h kg^−1^ per electron) than an Li-NiNC battery (46.42 A h kg^−1^ per electron) because of the larger formula weight of [28]Hex (F.W. = 1130.73 g/mol versus F.W. = 577.34 g/mol for NiNC). On the other hand, battery performances with slow charging conserved a consistent charge/discharge behavior with the redox potential curve as observed in CV. Both charge/discharge potential curves of the Li-[28]Hex battery were somewhat consistent with the respective oxidation and reduction curves of the CV. Interestingly, a specific chemical property was found, where the initial discharging pattern changed to another type. The former was rather simple: As battery performance progressed, the steps of two reduction potentials were more significant especially in the discharge steps, and decrease of the gap between the two potentials was observed (Appendix A).

### 2.2. Absroption Spectroscopy with Electrochemical Redox Reaction

Substantial interconversions were expected to be observed between Möbius and Hückel aromatic hexaphyrins. On the basis of the battery performance, the following have been hypothesized: (1) The structure of the initial [28]Hex does not fulfill a proper conformation to bear appropriate battery performances. (2) As the battery performances progressed, the [28]Hex molecule adjusts the molecular configuration, suitable to maintain easy interconversions between the different redox states. The oxidation state shifted readily to higher potentials, which resulted in a narrow gap of the redox potentials, whose outcome waved conformational changes from aromatic to antiaromatic. The interconversion probability between Möbius aromatic and Hückel antiaromatic [28]hexaphyrins were considered (Appendix A): The change in pattern of the HOMO–LUMO gap supported the hypothesis by density functional theory (DFT) calculations.

Nevertheless, further observation of color change in the electrochemical oxidation supported the oxidation from Hückel antiaromatic [28]Hex to Hückel aromatic [26]hex. Absorption changes of [28]Hex were measured in CH_2_Cl_2_. The electric current was set with 0.4 V and the time-dependent absorption changes were monitored (Figure 5). The spectrum colored in blue is the initial absorption spectrum of [28]Hex in CH_2_Cl_2_. As the electricity in the circuit flowed for the electrochemical oxidation, the absorption band in blue gradually changed to the band in green followed by the band in red at the end. Two isosbestic points were observed, elucidating a stepwise two-step redox process. 

The initial absorption spectrum was representative for [28]Hex. Furthermore, the absorption spectrum in red was representative of [26]hex. The result of electrochemical oxidation was particularly consistent with the oxidation, where the conversion of [28]Hex to [28]Hex radical cation proceeded and continued to the conversion of [28]Hex radical cation to [28]Hex di-cation. The final [28]Hex di-cation state was concluded to exhibit the same conjugation pathway with [26]hex, whose absorptions in CH_2_Cl_2_ were identical. The absorption spectral change as well as the solution color observation at the initial (blue) and final (purple) stages (Figure 6) supported the oxidation process of Hückel antiaromatic [28]Hex to Hückel aromatic [26]hex. The potential shift in charge/discharge performances (Appendix A) was concluded to be the preferential stabilization of the electrode materials, not the dynamics between Möbius aromatic and Hückel antiaromatic [28]hexaphyrins. Furthermore, there was no corresponding color appearance as well as no absorption spectrum for Möbius aromatic [28]Hex even with long term measurements.

### 2.3. Further Battery Performances

In long-term battery performances (100 cycle set of charge/discharge experiment with a fast charging rate of 1 mA), about 3 electrons were involved and the performing pattern was stabilized over the examination periods (Figure 7). Furthermore, when the charging rate was doubled (2 mA) to measure a 200 cycle battery capacity, the battery performance involved about 2 electrons, whose efficiency still reached nearly 95% (Figure 8).

Battery performances with 20% composite Li-[28]Hex batteries were then investigated. The increased composite amount of [28]Hex cracks its electrode discs easily in steps of compressing and drying. The furnished electrode discs were thicker, which decreased their surface area with the increased mass value, and the corresponding battery performances were not enhanced. However, a battery performance with a high charging speed (3 mA) was investigated in the 20% composite Li-[28]Hex battery. The exceedingly faster charging narrowed the window width to be 2~3.8 V, whose process involved only one electron. However, the Li-[28]Hex battery exhibited a stable charge/discharge performance over 1000 cycles (Appendix A). Improvement of the preparation method of active [28]Hex-electrodes to manipulate the fast charging is of interest for further research.

Finally, an overview of the Li-[28]Hex battery performance was delivered as the schematic pathway shown in Figure 9. The battery involved ideally 4 electrons per cyclic unit. Unfortunately, the enhanced rates gave incomplete results per cyclic unit, with only 2~3 electrons involved charge/discharge cycles. However, the performances were still reasonably stable over 200 cycles. However, the formation of 3D conductive networks can be an important issue [31,32]. This report was a result of the first study on the battery system having electrodes of expanded porphyrinoids, which was significantly challengeable. The corresponding concentration of carbon black was about 60%. The capacity of the pure carbon black battery exhibited around 20 mAh/g, which is not an ignorable error. As electrochemical impedance spectroscopy (EIS) plots (Appendix A) before and after 200 cycles of charge/discharge performances show significantly increased impedance, this suggests the formation of solid-state electrolyte interface (SEI) layers. Herein, it is planned to use about 50% weight of the organic material in further research. Further investigation on expanded porphyrinoid batteries is still ongoing in our research.

## 3. Materials and Methods

### 3.1. Preparation of Li-[28]hex Batteries

The compound for the active cathode of [28]Hex was prepared by following the literature method [33]. Freshly prepared [28]Hex was mixed with conductive carbon black (CB) and polyvinylidene fluoride ([28]Hex/PVDF/CB = 1:3:6 in mass ratio) and the mixture was blended carefully to get a fine powder. *N*-methyl-2-pyrrolidone was added to the powder, mixed, and pasted. The 10% and 20% composite electrodes were molded, pressed, and cut in a proper size (15 mm) for fitting into the coin cell. Approximately 38 and 34 mg weights in 1.5 cm of the diameter of the circles (21.5 mg/cm^2^ and 19.2 mg/cm^2^) were prepared as 10% and 20% composite cathodes, respectively. After drying for 2 h in a vacuum, Li-organic batteries were fabricated in coin cells. In a glove box, the completely dried [28]Hex-cathode, Li-foil anode, and porous polymer film separator were assembled with an electrolyte solution that was a composite of LiPF_6_ in a mixture of ethylene carbonate and diethyl carbonate. The 20% composite [28]Hex electrodes were prepared in the same process with a proper mass ratio of [28]Hex/PVDF/CB (1:1:3). Finally, individual battery behaviors of the Li-organic electrodes were examined in optimized and programed conditions.

### 3.2. Optimization of the Current Window for Best Charge/Discharge Processes

Cyclic voltammetry was investigated with a Li-[28]Hex coin-cell battery. The initial range (over 4 V) was more comprehensive than that shown in Figure 3, and caused damage to the battery due to the over-oxidation of Li, thus the current window was fixed in 2–4 V. Two to three batteries were scheduled in the same conditions, and charge/discharge behaviors were measured simultaneously. Areal loading rates affected the battery performance due to poor diffusion rates of the electrolyte with the matrix material. Slow charging of 0.057 mA/cm^2^ or 0.11 mA/cm^2^ exhibited chemical properties significantly. Increased charging speed (0.28, 0.57, and 1.14 mA/cm^2^) rejected the preservation of the specific curve for the [28]Hex electrode. However, raising the speed permitted long period battery performances.

## 4. Conclusions

In conclusion, lithium-collaborating organic batteries (Li-[28]hexs) were investigated with [28]hexaphyrin(1.1.1.1.1.1) active electrode material. Each hexaphyrin molecule of the [28]Hex cathode involved four electrons performing typical charge/discharge processes of Li-[28]Hex batteries. Li-[28]Hex batteries performed stable charge/discharge processes over long periods. Oxidation of Hückel antiaromatic [28]hexaphyrins toward Hückel aromatic [26]hexaphyrins were incorporated with the charge/discharge performances of the Li-[28]Hex batteries. Further investigations on such type of organic batteries using porphyrinoids as well as consistently proficient organic molecules suitable for the active electrode materials are still underway in our research.

## Figures and Tables

**Figure 1 molecules-24-02433-f001:**
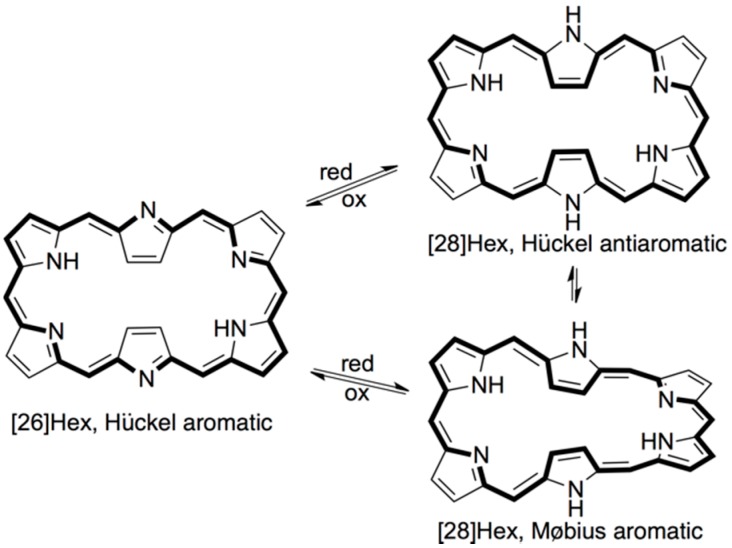
Conformation dynamics in accordance with redox reaction.

**Figure 2 molecules-24-02433-f002:**
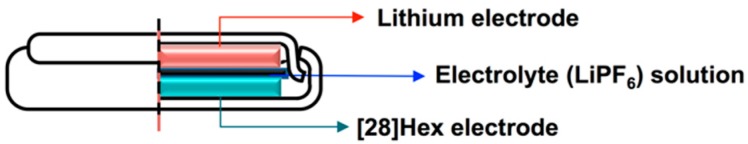
Schematic picture of Li-[28]Hex battery.

**Figure 3 molecules-24-02433-f003:**
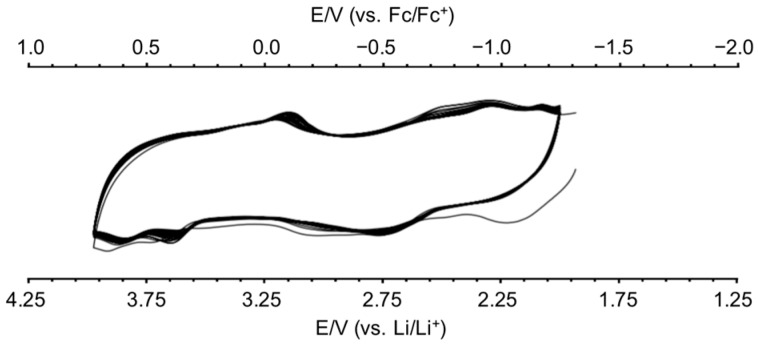
Cyclic voltammogram of Li-[28]Hex battery.

**Figure 4 molecules-24-02433-f004:**
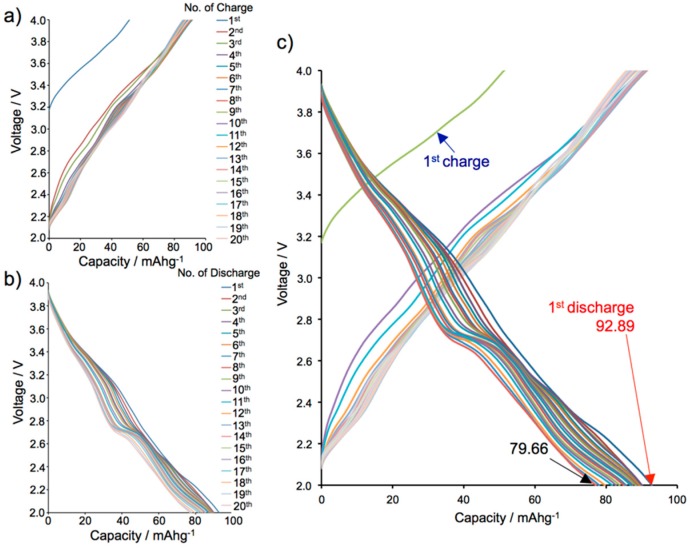
Charge/discharge performances of Li-[28]Hex battery: range = 2~4 V, number of cycles = 20, and operation current = 0.2 mA. (**a**), (**b**), and (**c**) show charge performances, discharge performances, and overall charge and discharge performances, respectively.

**Figure 5 molecules-24-02433-f005:**
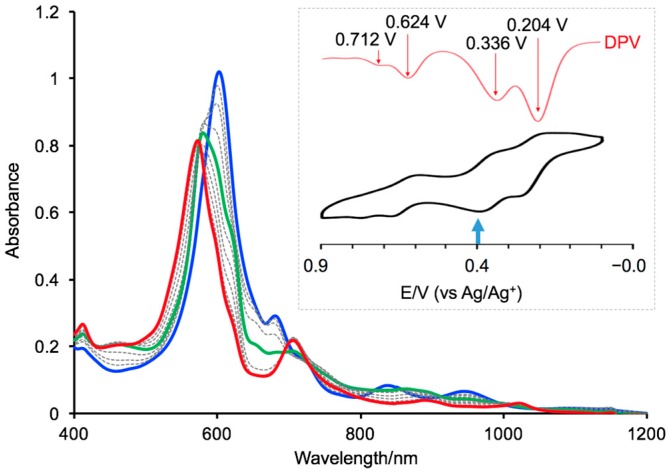
Absorption spectral changes of a CH_2_Cl_2_ solution of [28]Hex containing 0.1 M nBu_4_NPF_6_, upon the electrochemical oxidation (set with 0.4 V): The absorption spectral changes were monitored with linear sweep voltammetry set in 0.4 V for the initial and final potentials (working electrode: Pt mesh, counter electrode: Pt, reference cell: Ag/AgCl_4_). Inner graph represents oxidation potentials of [28]Hex; the graph in black is for cyclic voltammogram (scan rate = 0.2 V/s) and the graph in red is for differential pulse voltammogram, respectively (working electrode: Pt, counter electrode: Pt, reference cell: Ag/AgCl_4_).

**Figure 6 molecules-24-02433-f006:**
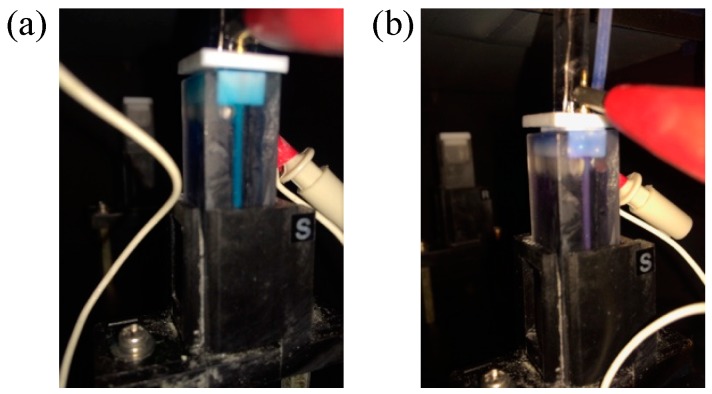
Color change by cyclic voltammetry-UV (CV-UV) measurement: (**a**) Before and (**b**) after the measurement.

**Figure 7 molecules-24-02433-f007:**
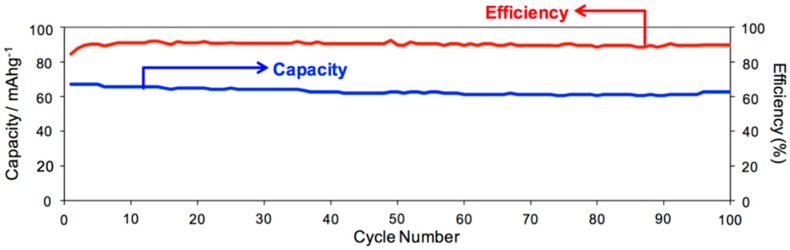
Capacity and efficiency of Li-[28]Hex battery over 100 cycle measurement: window’s width = 2~4 V, operation current = 1 mA.

**Figure 8 molecules-24-02433-f008:**
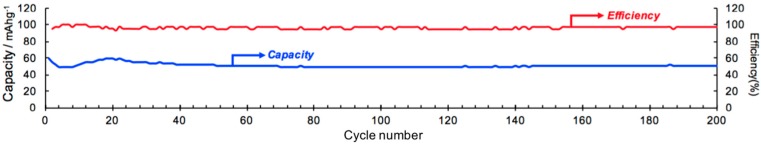
Capacity and efficiency of Li-[28]Hex battery over 200 cycle measurement: window’s width = 2~4 V, operation current = 2 mA.

**Figure 9 molecules-24-02433-f009:**
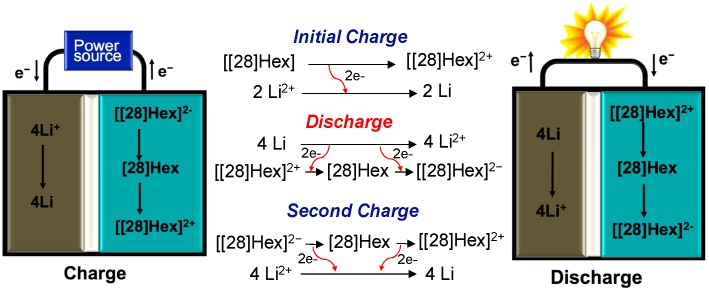
Summary of the charge/discharge reactions for Li-[28]Hex battery.

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
