# Peer review of "Exploration of Li-Organic Batteries Using Hexaphyrin as an Active Cathode Material"

_molecules, 2019, doi:10.3390/molecules24132433_

Reviewer 1 Report

The paper describes the application of [28]hexaphyrin(1.1.1.1.1.1) as a cathode material for Li-ion batteries. Despite the importance of the topic, clarity of the data and their presentation should be improved. 1) What was the loading of the active material in the cell (g/cm2)? 2) The charge current units are not consistent trough the text: p 3 l127 "mA/s" (what is "s" if charge is in a constant current mode?)  p 3 l135 "mA" (it is not clear if it is normalized per cm2, g or not? If not normalized than it should be for the consistence of the data)  p 7 l 179 the same current is called "mA/sec". Absolutely unclear what authors meant by this normalization. 3) What is "steps = 0.2 mA/sec." p 5 l 151? Again different name for a charging current? The same in the caption for figs. 8 and 9 and S4. 4) Figure 1 has no Li+ participating in reactions, while the scheme of the same reactions on Fig 10 involves Li+. Please, clarify. 5) Caption of Fig 6 is not informative, please, include more detailes from the text. 6) Why CVs have shape close to rectangular with small Red/Ox peaks? May the carbon additive have noticeable double layer capacity overlapping the redox processes of [28]hex? Please, show the background CV and charge-discharge curve (for electrodes with the same carbon content but without [28]hex).

Author Response

Q1) What was the loading of the active material in the cell (g/cm2)?

To give the answer, we added the following information in the revision (lines 178-179).  "Approximately 38 and 34 mg weights in 1.5 cm of the diameter of the circles (21.5 mg/cmand 19.2 mg/cm2) were prepared as 10% and 20% composite cathodes, respectively."

Q2) The charge current units are not consistent through the text: p 3 l127 "mA/s" (what is "s" if charge is in a constant current mode?)  p 3 l135 "mA" (it is not clear if it is normalized per cm2, g or not? If not normalized than it should be for the consistence of the data) p 7 l 179 the same current is called "mA/sec". Absolutely unclear what authors meant by this normalization.

I am sorry for the typing mistake. At the moment, I was teaching students ‘kinetics, where reaction rates were much concerned.  The "second" was insensitively added in the unit of charging rate.  Since ampere already includes the term of unit time (A = C/s), the second is duplicated.  The units were corrected. 

Q3) What is "steps = 0.2 mA/sec." p 5 l 151? Again different name for a charging current? The same in the caption for figs. 8 and 9 and S4.

All names for charging current shown in the manuscript including the captions of figures were corrected to be consistent.

Q4) Figure 1 has no Li+ participating in reactions, while the scheme of the same reactions on Fig 10 involves Li+. Please, clarify.

Figure 1 only represents magnetic behavior on [28]hexaphyrin molecule which exhibits dynamics between aromaticity and antiaromaticity, which we aimed to observe, but didn’t observe.

The possibility of Li+ metallation was denied when the color of the hexaphyrin electrode was checked: the Li-[28]hex battery that was disassembled after charge/discharge processes, exhibited a typical color (violet) of [26]hexaphyrin.

Q5) Caption of Fig 6 is not informative, please, include more details from the text.

The details of the experiment were included in the caption of Figure 6. Furthermore, since Figure 5 was moved from the manuscript to the supporting information, each figure number was corrected.

Q6) Why CVs have a shape close to rectangular with small Red/Ox peaks? May the carbon additive have noticeable double layer capacity overlapping the redox processes of [28]hex? Please, show the background CV and charge-discharge curve (for electrodes with the same carbon content but without [28]hex).

The background CV that was measured with only consistent carbon black was added in the supporting information (Figure S5).

We appreciate the valuable comments and suggestions of the reviewers. Hopefully, this manuscript is now suitable for the publication to your journal.

Reviewer 2 Report

1.In the preparation of cathode material, carbon black concentration is around 60% for both 10% and 20% composites. Even though high concentration carbon black is not favorable in practice, it is important in a fundamental study to ensure electronic conductivity by forming 3D conductive network due to percolation theory via adding enough carbon black, the strategy and mechanism of which have been detailed in the reference papers of "Electrochimica Acta, 161, 322-328 (2015)" and "J. Electrochem. Soc. 162, A1601-A1609 (2015)". The authors should provide discussion on the necessity of using high loading carbon black and cite these two important papers. Besides, the authors are recommended to provide a control group of pure carbon black (as cathode) in a battery to reveal the capacity contribution of carbon black. It is known that high concentration carbon black could lead electrical double layer effect, which was also evaluated in these two reference papers.

2.The authors should provide discussion on the electrochemical reactions corresponding to the reduction/oxidation peaks observed in Figure 3. Besides, further comparison of plateaus (Figure 4) to redox reactions (Figure 3) is needed. Why the plateau at 2.7 V becomes obvious after cycling?

3. The authors might want to perform electrochemical impedance spectroscopy (EIS) before and after cycling to evaluate electrode/electrolyte interphase stability, revealing the cycling stability of the active material.

Author Response

Q1. In the preparation of cathode material, carbon black concentration is around 60% for both 10% and 20% composites. Even though high concentration carbon black is not favorable in practice, it is important in a fundamental study to ensure electronic conductivity by forming 3D conductive network due to percolation theory via adding enough carbon black, the strategy and mechanism of which have been detailed in the reference papers of "Electrochimica Acta, 161, 322-328 (2015)" and "J. Electrochem. Soc. 162, A1601-A1609 (2015)". The authors should provide a discussion on the necessity of using high loading carbon black and cite these two important papers. Besides, the authors are recommended to provide a control group of pure carbon black (as a cathode) in a battery to reveal the capacity contribution of carbon black. It is known that high concentration carbon black could lead to electrical double layer effect, which was also evaluated in these two reference papers.

Yes, the reviewer’s comment is correct and considerable.  The formation of 3D conductive network can be an important issue. This manuscript reports a result of the first study on the battery system having electrodes of expanded porphyrinoids, which was significantly challengeable. We keep trying to investigate the better battery systems and improve the method for preparing the enhanced component ratios of active cathode molecules versus that of carbon black.  We prepared batteries having pure carbon black without [28]hexaphyrin and then measured the battery performance. The corresponding concentration of carbon black was about 60%. The capacity of the pure carbon black battery exhibited around 20 mAh/g which is not an ignorable error. As EIS plots (those were given in Figure S6) before and after 200 cycles of charge/discharge performances show significantly increased impedance, this suggests the formation of solid-state electrolyte interface (SEI)layers. We appreciate your comment.  We plan to use about 50% weight of the organic material in further research.  To increase the component ratio, we have to use new fabrication methods, which will be distinct from the earlier fabrication.  The papers recommended by the reviewer were cited in the references (references 31 and 32).

Q2.The authors should provide a discussion on the electrochemical reactions corresponding to the reduction/oxidation peaks observed in Figure 3. Besides, further comparison of plateaus (Figure 4) to redox reactions (Figure 3) is needed. Why the plateau at 2.7 V becomes obvious after cycling?

As described in the manuscript, we first expected to observe substantial conversion between Hückel antiaromatic [28]hexaphyrin and Möbiusaromatic [28]hexaphyrin, which seemed to show agreement with the data of the DFT calculation (Figure S2). The observation of the plateau at 2.7 V probably is the sign for the aromatic change of [28]hexaphyrin. However, absorption behaviors along the electrochemical oxidation showed the charge/discharge happened simply between Hückel antiaromatic [28]hexaphyrin and Hückelaromatic [26]hexaphyrin. As a result, the potential shift in charge/discharge performances was concluded to be the preferential stabilization of the electrode materials, not the dynamics between Möbius aromatic and Hückel antiaromatic [28]hexaphyrins.

Q3. The authors might want to perform electrochemical impedance spectroscopy (EIS) before and after cycling to evaluate electrode/electrolyte interphase stability, revealing the cycling stability of the active material.

The EIS plots before and after battery cycles were added in Figure S6.

Overall, we appreciate the valuable comments and suggestions of the reviewers.  Hopefully, this manuscript is now suitable for the publication to your journal.